# Deep Soil Water Content and Forage Production in a Tropical Agroforestry System

**Marcos Vinicius Mansano Sarto** [1,*,†] , **Wander Luis Barbosa Borges** [2] , **Doglas Bassegio** [3] , **Márcio Renato Nunes** [4] , **Charles W. Rice** [1] **and Ciro Antonio Rosolem** [5]

1  Department of Agronomy, Kansas State University, Manhattan, KS 66506, USA; rice@ksu.edu
2  Advanced Research Center of Rubber Tree and Agroforestry Systems, Agronomic Institute/IAC, Votuporanga 15500-970, SP, Brazil; wanderborge@iac.sp.gov.br
3  Department of Energy Engineering in Agriculture, State University of Western Paraná, Cascavel 85816-360, PR, Brazil; doglas.bassegio@unioeste.br
4  Soil and Water Sciences Department, University of Florida, Gainesville, FL 32611, USA; marcionunes@gmail.com
5  Department of Crop Science, São Paulo State University, UNESP, Rua José Barbosa de Barros, 1780, Botucatu 18610-307, SP, Brazil; rosolem@fca.unesp.br
*  Correspondence: sarto@ksu.edu; Tel.: +1-(785)-317-4619
†  Present address: Throckmorton Plant Sciences Center, Kansas State University, 1712 Claflin Road, Manhattan, KS 66502, USA.

**Abstract:** Agroforestry systems integrating tree and forage growth are important for maintaining soil health but may change the soil's physical-hydric properties. Our goal was to investigate the impact of introducing *Eucalyptus* trees into a pasture on the soil water content throughout the soil profile. The study was conducted in a 6-year-old agroforestry system where two species of *Eucalyptus* were introduced into a palisade grass pasture. Soil moisture was sampled at 0.0 (planting row), 2.0, 4.0, and 6.0 m (midpoint between tree rows) from the *Eucalyptus* tree rows. A monoculture palisade grass pasture was used as a control. The soil water content down to a depth of 50 cm was lowest in the tree row and increased with distance from the trees. In the *Eucalyptus* row, the soil water content in the 0–50 cm layer was lower than in the monoculture pasture. Agroforestry systems decreased the water content in the superficial layers of the soil in the rainy months; in the dry season, the soil water contents in all layers were similar between the *Eucalyptus* inter-rows. In most seasons, the agroforest systems reduced the forage production close to the *Eucalyptus* tree rows, up to 2 m from the trees, likely due to the soil water content decrease. Overall, this study showed that in tropical regions with sandy soils, the grass and trees' competition must be considered when establishing integrated agroforestry systems in order to maximize the advantages and benefits of the diversified agroecosystem.

**Keywords:** *Eucalyptus*; *Urochloa brizantha*; soil moisture; water uptake

## 1. Introduction

Agroforestry improves soil quality, carbon sequestration, and water quality in cropping systems [1–3] and reduces runoff and nutrient losses [4]. Furthermore, this land-use system promotes the efficient use of resources (e.g., nutrients, light, and water) compared with traditional soil and crop management practices [5]. In general, compared with intensive farming systems, the soil quality under agroforestry systems tends to be more similar to that under natural vegetation [6,7]. In addition, agroforestry increases vertical and horizontal root growth compared with annual cropping systems, leading to increased porosity and pore continuity in the subsurface horizons [8], as well as greater water infiltration and storage [9].

The integration of components in the agroforest systems can result in positive, neutral, or negative interactions, depending on the outcome of the competition [10]. An important

factor limiting the adoption of agroforestry systems by farmers is the competition between trees and crops for solar radiation, water, and nutrients [11,12]. The soil water content in agroforestry systems is influenced by microclimatic conditions such as solar radiation, wind speed, and rainfall [13]. In addition, by reducing solar radiation, shade affects microclimatic variables, including air and soil temperatures, relative humidity, soil moisture [14,15], and evapotranspiration [16], thereby influencing crop growth [17]. Non-uniform soil variations and competition for water have been observed in agroforestry systems in which *Eucalyptus* is integrated with annual crops and pastures [18]. The competition for water is also affected by root distribution [19] and decreases in crop or forage root length due to the fact that proximity to tree rows could result in a reduced capacity to compete for resources [20]. If agroforest systems are not properly scaled and managed, the potential advantages and benefits provided by the addition of the trees can be negated due to the competition for water or radiation [10]. Some studies suggest that wider tree spacing within rows contributes to maintaining forage production [21]. Gomes et al. [22] found that morphologic composition and sward characteristics were affected up to 3 m from the trees.

The interaction between the trees and pasture may vary between the dry and rainy seasons. In dry environments, the soil water content tends to decrease under vegetation due to transpiration, increasing the spatial variability of the soil water content. The difference in water use between the trees and pasture is smaller when the water supply is limited [19]. Given their capacity to improve water infiltration and storage in the soil [23], agroforestry systems may be an important strategy to mitigate the effects of climate change on Brazilian pasturelands [24]. The present study hypothesized that introducing *Eucalyptus* trees in an agroforestry system would change the soil water content and lead to greater competition between trees and pasture, especially under dry conditions. Accordingly, the aim of this study was to evaluate changes in the soil water content in the linear transect perpendicular to *Eucalyptus* rows grown in an agroforestry system with a pasture compared with a monoculture pasture.

## 2. Materials and Methods

### 2.1. Characterization of the Experimental Area

Samples were taken from a 6-year-old agroforestry system experiment in Votuporanga, State of São Paulo, Brazil (50°04′ W, 20°28′ S; 450 m above sea level). The experiment started in 2009 on a degraded pasture (Figure 1).

The region has a tropical climate categorized as Cwa according to Köppen's classification, with mild, dry winters and hot, rainy summers [25]. The annual average maximum and minimum temperatures are 30 °C and 18 °C, respectively, with annual rainfall between 1100 and 1500 mm. Rainfall and maximum and minimum temperatures were measured throughout the experiment (Figure 1D).

The soil of the experimental area was classified as Arenic Hapludult [26]. Table 1 shows the soil chemical [27] and physical (i.e., bulk density [28] and particle size distribution [29]) characteristics in the experimental area within the soil profile (0–100 cm) at the beginning of the experiment in 2009.

### 2.2. Experimental Design and Treatments

The experimental design was completely randomized with six replicates from October 2013 to June 2015. Four sampling locations were selected at 0.0 (planting row), 2.0, 4.0, and 6.0 m (midpoint between rows) from the *Eucalyptus* tree rows (Figure 1B), at depths of 0–10, 10–20, 20–30, 30–40, 40–50, 50–60, 60–70, 70–80, 80–90, and 90–100 cm. In addition, a monoculture pasture on the same soil and managed in the same way as the agroforestry systems was used as reference (Figure 1C). The plots had an area of approximately 1.0 ha.

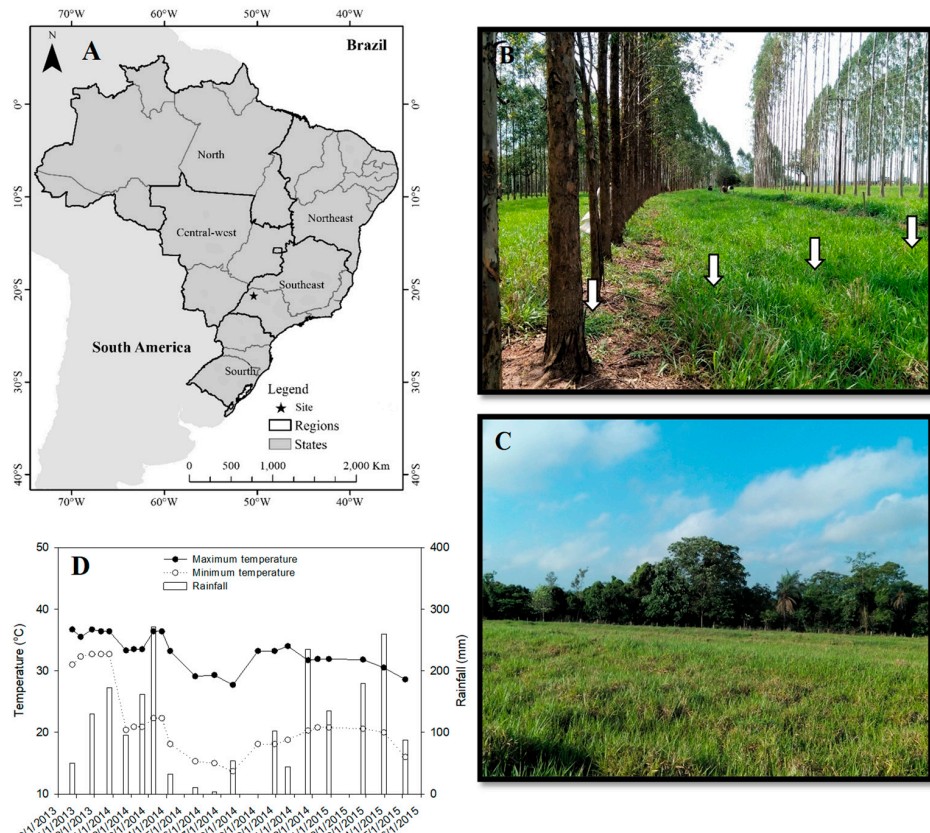

**Figure 1.** (**A**) The experimental area in São Paulo state, Brazil. (**B**) Sampling locations: 0.0 (planting row), 2.0, 4.0, and 6.0 m from the *Eucalyptus* trees. (**C**) Monoculture pasture. (**D**) Rainfall and maximum and minimum temperatures in the experimental area, 2013–2015.

**Table 1.** Chemical and physical soil characteristics of the studied Arenic Hapludult prior to the start of the experiment in 2009.

| Soil Depth | P (Resin) | SOM [a] | pHCaCl$_2$ | K$^+$ | Ca$^{2+}$ | Mg$^{2+}$ | H + Al | Al$^{3+}$ | BS [b] |
|---|---|---|---|---|---|---|---|---|---|
| cm | mg dm$^{-3}$ | g kg$^{-1}$ | | | | mmol$_c$ dm$^{-3}$ | | | % |
| 0–5 | 2.89 | 12.2 | 4.9 | 1.57 | 19.4 | 13.7 | 24 | 0.36 | 55 |
| 5–10 | 0.78 | 11.7 | 4.7 | 1.32 | 7.0 | 4.1 | 31 | 1.12 | 29 |
| 10–20 | 0.33 | 11.5 | 4.9 | 1.04 | 6.2 | 3.1 | 32 | 1.39 | 25 |
| 20–40 | 0.35 | 10.4 | 4.7 | 0.71 | 7.5 | 3.3 | 31 | 1.28 | 27 |
| 40–60 | 0.30 | 9.8 | 5.1 | 0.88 | 8.0 | 3.3 | 22 | 2.47 | 35 |
| 60–80 | 0.44 | 9.8 | 5.2 | 0.80 | 8.2 | 3.5 | 21 | 1.69 | 26 |
| 80–100 | 0.44 | 7.2 | 5.3 | 0.57 | 9.1 | 3.8 | 19 | 1.40 | 41 |

| Soil depth | Sandy | Silt | Clay | Ds [c] | Texture |
|---|---|---|---|---|---|
| cm | | g kg$^{-1}$ | | Mg m$^{-3}$ | |
| 0–5 | 816 | 102 | 82 | 1.58 | Loamy sand |
| 5–10 | 820 | 73 | 107 | 1.60 | Loamy sand |
| 10–20 | 809 | 68 | 123 | 1.60 | Loamy sand |
| 20–40 | 783 | 75 | 142 | 1.60 | Sandy loam |
| 40–60 | 776 | 69 | 155 | 1.55 | Sandy loam |
| 60–80 | 746 | 62 | 192 | 1.53 | Sandy loam |
| 80–100 | 728 | 64 | 208 | 1.47 | Sandy loam |

[a] Soil organic matter; [b] Base saturation; [c] Bulk density.

### 2.3. Site History

The experiment was carried out in a 10-year degraded pasture of approximately 6 ha. The area was prepared by plowing and disking, and 2000 kg ha$^{-1}$ of limestone was applied

to raise the soil base saturation to 60%. In October 2009, 200 kg P ha$^{-1}$ of phosphorus was applied using reactive rock phos-phate. *Eucalyptus* [hybrid *E. urograndis* H–13 (*Eucalyptus urophylla* × *E. grandis*) and hybrid *E. grancam* 1277 (*E. grandis* × *E. camaldulensis*)] were planted east–west with a 12.0 m distance between rows and 2.0 m between plants using 3-month-old seedlings at a density of 370 plants ha$^{-1}$. Palisade grass (*Urochloa brizantha* cv. Marandu) was sown 8.0 cm deep at a density of 7 kg ha$^{-1}$ (pure live seed = 76%) using a no-till drill at a row spacing of 0.34 m.

Two years after planting, the *Eucalyptus* trees were trimmed to a height of 12.0 m, and 2/3 of the crown was removed to allow more light to reach the pasture. Nitrogen was broadcast on the pasture in both the integrated system and monoculture pasture at 90 kg ha$^{-1}$ N as urea at the beginning of the period of higher rainfall (October).

In September 2011, freshly weaned beef cattle with an average weight of 174 kg were introduced (4.4 head ha$^{-1}$) in the agroforestry system and monoculture pasture until slaughter in June 2014, at which time they had an average weight of 492 kg. In September 2014, a new group of freshly weaned beef cattle with an average weight of 179 kg was introduced at a stocking rate of 3.2 head ha$^{-1}$ and grazed until the end of the experiment, when their average weight was 342 kg.

### 2.4. Soil Water Content

Soil volumetric water content was measured from October 2013 to June 2015 with a capacitance probe (model Diviner 2000$^®$, Sentek Pty Ltd., Stepney South, Australia) at 0.0 (planting row), 2.0, 4.0, and 6.0 m from the *Eucalyptus* tree row and in the monoculture pasture. On average, the measuring interval was every 25 days. The measurements were executed on the same day, between 9 and 11 a.m. In total, 8820 readings were acquired.

The soil water content was measured every 0.1 m to a depth of 1.0 m, at the area with the greatest presence of roots. From 1.0 m, the fine roots decreased gradually [30]. At each sampling location, the portable probe was inserted into each of two previously installed access PVC tubes, and the two readings were combined into one reading per location per plot. The access tubes were placed into 5.0-cm-diameter soil boreholes, and a soil-water mixture was poured along the tube walls to improve contact with the bulk soil. The sensor exhibited measurement precision better than 0.01 m$^3$ m$^{-3}$.

### 2.5. Forage Production

The forage biomass was evaluated by cutting the forage within a 0.5 × 0.5 m (0.25 m$^2$) frame. The samples were taken after the rainy season (April) and the dry season (October) in the years 2014 and 2015. We sampled four subsamples randomly from each plot and combined them into one composite forage sample per plot. After cutting, the samples were oven-dried at 60 °C for 72 h and weighed to obtain the dry matter yield. The forage production was extrapolated to Mg ha$^{-1}$.

### 2.6. Statistical Analysis

The data were checked for normality (Shapiro–Wilk test), the independence of residuals, and the variance of homogeneity (Levene's test). All statistical analyses were performed by Sisvar$^®$ software (Statistical Analysis Software, UFLa, Lavras, MG, Brazil). The dataset of soil water was submitted to ANOVA for each month and means were compared by Fisher's protected LSD test ($p < 0.05$). The forage production was statistically significant at $p < 0.05$, by Fisher's protected LSD test ($p < 0.05$).

## 3. Results

### 3.1. Soil Water Content

From October to April 2013/2014 and from October to April 2014/2015 (rainy seasons), the soil water content was significantly ($p < 0.05$) lowest close to the *Eucalyptus* tree rows, i.e., at 0.0 and 2.0 m in the 0–10 cm (Figure 2A), 10–20 cm (Figure 2B), 20–30 cm (Figure 2C), 30–40 cm (Figure 2D), and 40–50 cm (Figure 2E) soil layers. In these months, especially at

the 0–10 and 10–20 cm soil layers, the soil water content was significantly higher ($p < 0.05$) in the monoculture pasture compared to the soil water content close to the *Eucalyptus* tree rows (i.e., at 0.0 and 2.0 m).

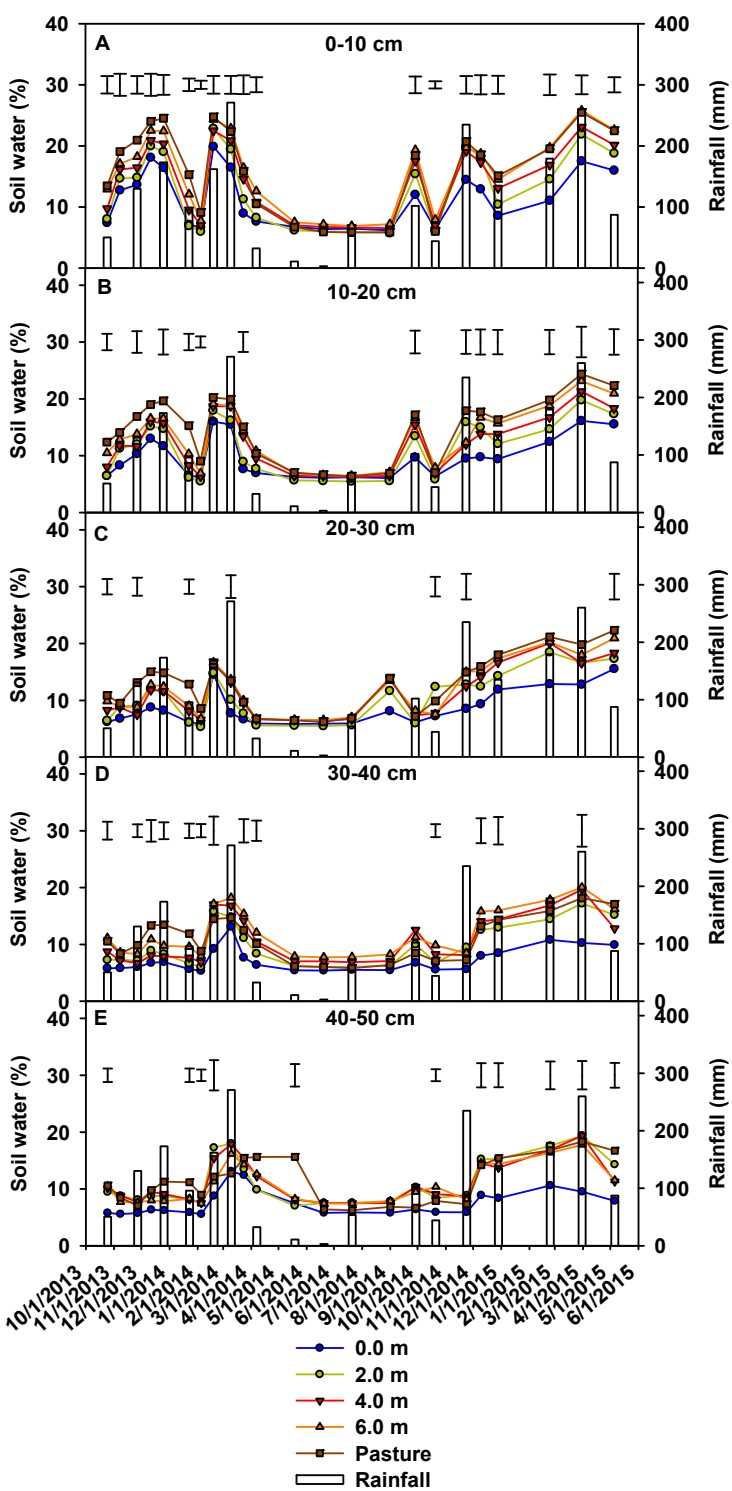

**Figure 2.** Average soil water content at soil depths of (**A**) 0–10 cm, (**B**) 10–20 cm, (**C**) 20–30 cm, (**D**) 30–40, and (**E**) 40–50 cm as a function of sampling location: 0.0 (planting row), 2.0, 4.0, and 6.0 m from the *Eucalyptus* tree rows and in monoculture pasture. Bars indicate LSD (0.05) values for dates with significant differences between sampling locations.

From April to September (dry season), the soil water content in the *Eucalyptus* inter-rows (i.e., at 0.0, 2.0, 4.0, and 6.0 m) was significantly similar ($p > 0.05$) to that in the monoculture pasture in all layers (Figures 2 and 3). In these months, the minimum average soil water content was 7.6% in soil layers deeper than 50–60 cm (Figure 2) and 8.2% in the 0–10 cm layer (Figure 2A).

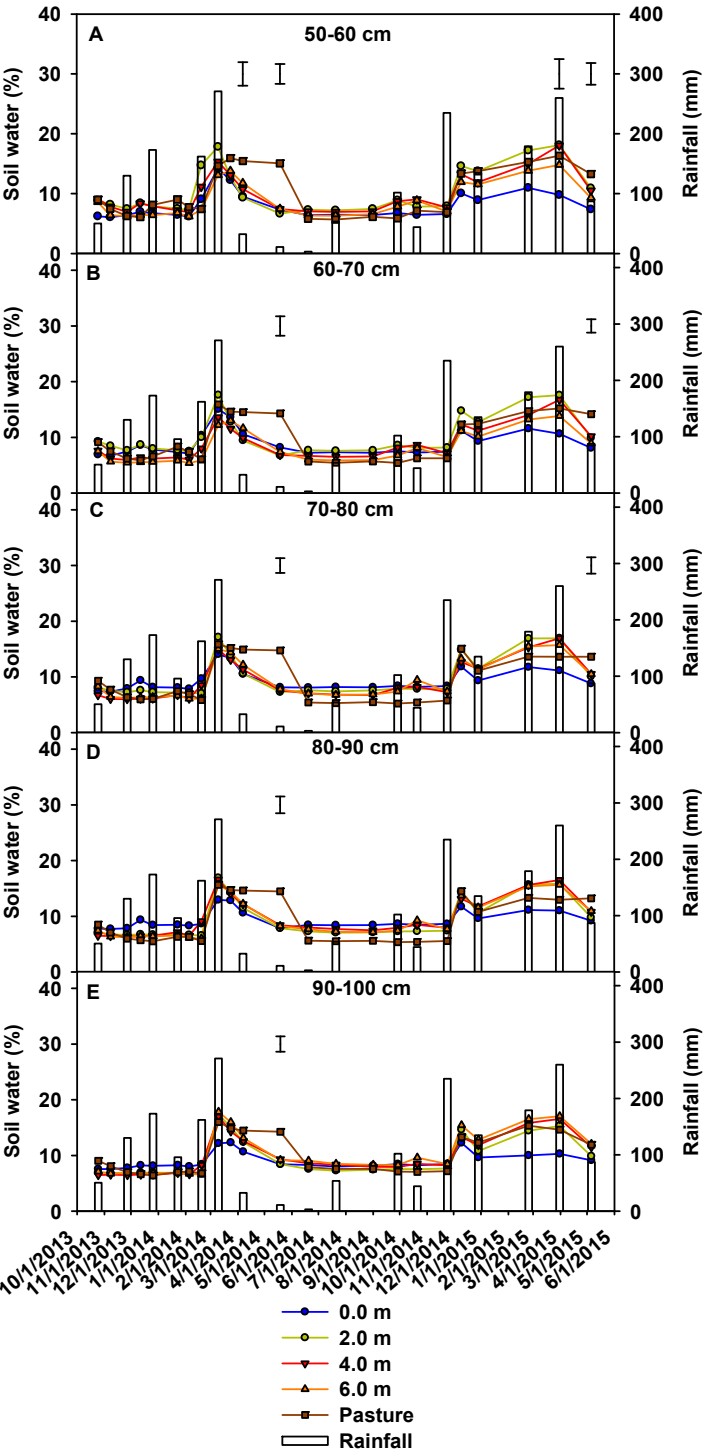

**Figure 3.** Average soil water content at soil depths of (**A**) 50–60 cm, (**B**) 60–70 cm, (**C**) 70–80 cm, (**D**) 80–90, and (**E**) 90–100 cm as a function of sampling location: 0.0 (planting row), 2.0, 4.0, and 6.0 m from the *Eucalyptus* tree rows and in monoculture pasture. Bars indicate LSD (0.05) values for dates with significant differences between sampling locations.

Except for the dry months (April to September 2014), at 4.0 and 6.0 m from the tree rows, the soil water contents in the 0–10 and 10–20 cm layers were significantly similar ($p > 0.05$) to that in the monoculture pasture (Figure 2A).

In May 2014 (dry season), at depths greater than 40–50 cm, the soil water content was higher significantly ($p < 0.05$) in the monoculture pasture than in the *Eucalyptus* inter-rows (i.e., 0.0, 2.0, 4.0, and 6.0 m; Figures 2E and 3A–E). In the rainy seasons, the variation in the soil water content at depths greater than 50–60 cm was low ($p < 0.05$) in the *Eucalyptus* transect (0.0, 2.0, 4.0, and 6.0 m) and monoculture pasture (Figures 2 and 3).

In general, the soil water content increased significantly ($p < 0.05$) with distance from the trees in soil layers up to a depth of 30–40 cm (Figure 4). Averaged across months, the soil water content in the 30–40 cm layer was 7.2% lower ($p < 0.05$) under the *Eucalyptus* tree row than in the monoculture pasture (Figure 4).

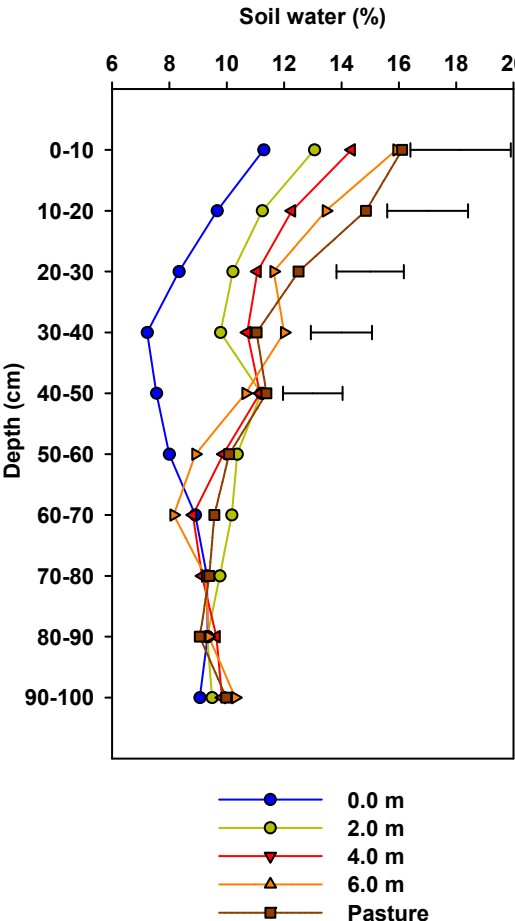

**Figure 4.** Average soil water content in the 2013–2015 period as a function of sampling location: 0.0 (planting row), 2.0, 4.0, and 6.0 m from the *Eucalyptus* tree rows and in monoculture pasture. The bars indicate LSD (0.05) values for locations with significant differences.

### 3.2. Forage Biomass

The forage biomass was affected significantly ($p < 0.05$) by the addition of *Eucalyptus* in the system, except across sampling in October 2014 (Figure 5). In October 2014, after the dry season, the forage biomass ranged from 1.6 to 2.2 Mg ha$^{-1}$, but there were no differences ($p < 0.05$) between the agroforestry systems and the pasture (Figure 2B).

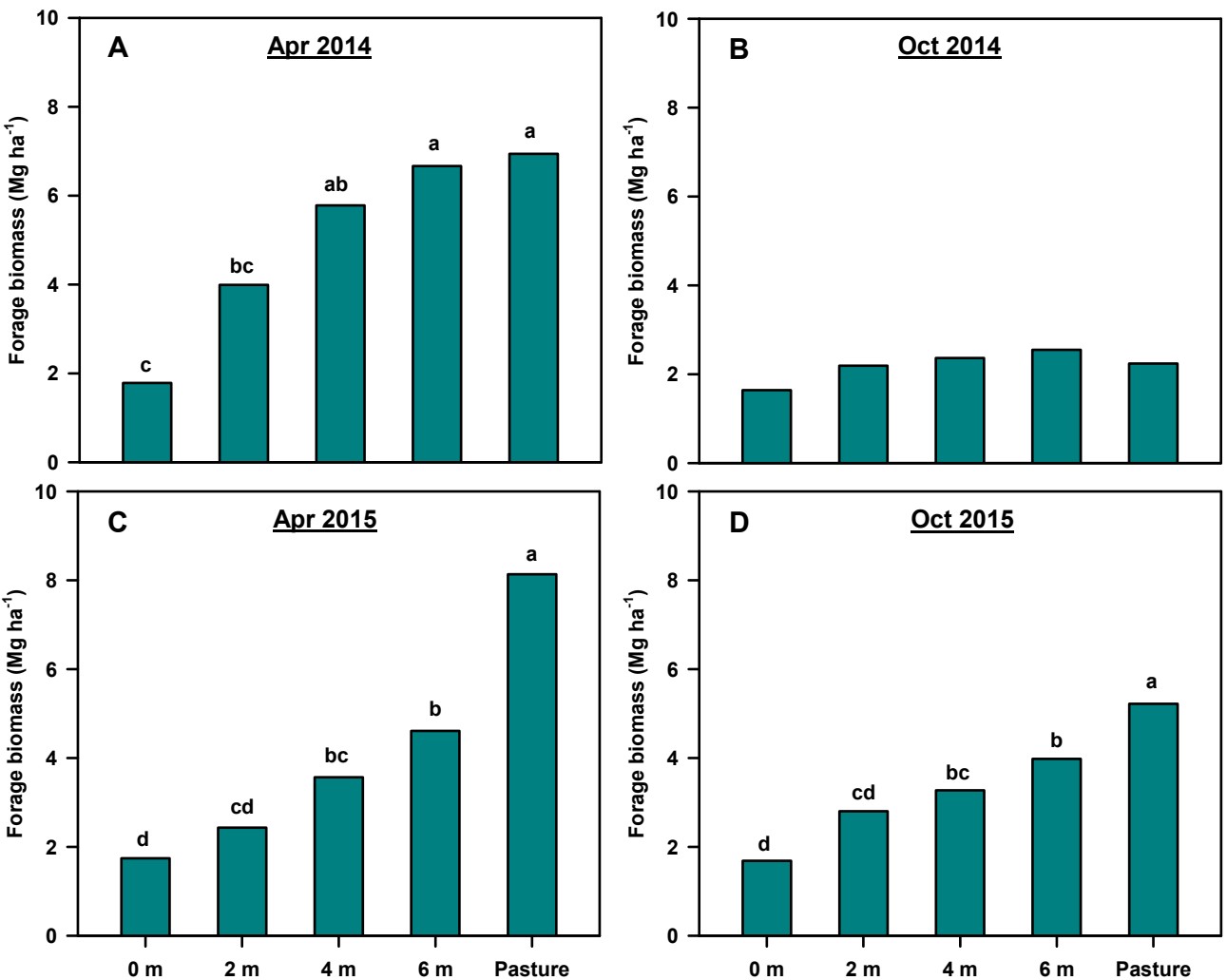

**Figure 5.** Production of dry matter biomass of the forage in (**A**) April and (**B**) October in 2014, and (**C**) April and (**D**) October in 2015 in the agroforestry systems and pasture. Means for forage in each season followed by the same letters do not differ significantly by the *t*-test (LSD), 5% probability.

In April 2015 and October 2015, the forage biomass was significantly lower ($p < 0.05$) in the agroforestry systems (3.0 Mg ha$^{-1}$) compared to the pasture (6.7 Mg ha$^{-1}$; Figure 5C,D). In addition, across sampling times, the forage biomass was 36% lower in the agroforestry systems close to the trees up to 2.0 m (2.9 Mg ha$^{-1}$) from the trees compared to 6.0 m (4.5 Mg ha$^{-1}$).

## 4. Discussion

In general, the soil water content was lower in the *Eucalyptus* tree rows than in the monoculture pasture throughout the year, with greater differences in months with higher rainfall (Figures 2 and 3). This difference in the soil water content reflects the interspecific competition. Competition for light and/or water has been reported to decrease the yields of some species [31,32]. The tropical sandy soil examined in the present study has low water storage capacity, increasing the importance of the competition for water. The lower soil water content within the *Eucalyptus* tree rows reflects a higher water uptake by plants due to the concentration of fine roots of the *Eucalyptus* [33]. Radersma and Ong [34] observed decreases in the soil water content of 2–7% (vol. basis) near fast-growing species such as *E. grandis* and *Grevillea robusta* and linked these decreases to greater water use by *Eucalyptus* and Grevillea compared with Cedrella and Markhami in a Rhodic Acrudox with 58% clay.

Prasad et al. [35] also attributed low water content at depths of up to 40 cm in the proximity of tree rows to water absorption by the trees.

In general, the soil water content increased with distance from the trees (Figures 1 and 4), consistent with previous reports [3,36]. Greater water uptake near trees was also observed by Pezzopane et al. [14]. Siriri et al. [23] showed that Calliandra tree species reduced the yields of beans and maize up to 6.0 m from the Calliandra row, indicating that roots of this species can extend through lateral and vertical distances and compete with the companion crop. Pezzopane et al. [37] observed lower soil moisture at distances of 0.5–1.0 m from trees, most likely due to higher water uptake by the tree roots and increased drainage due to improvements in soil structure from the tree root systems [38].

In the dry season, from April to September, the soil water content did not differ between the *Eucalyptus* row and the different distances (Figures 2 and 3). Bosi et al. [13] reported lower soil water availability in the *Eucalyptus* row than at a distance of 7.50 m during dry spells. In dry environments, the water competition from trees generally results in low yields of the companion crop [39]. According to Wang et al. [40], the soil water content under vegetation tends to decrease in drier periods due to transpiration, increasing the spatial variability of the soil water content. The different effects of tree species on crop performance were less obvious in seasons with low rainfall, suggesting that differences in water use among tree species are smaller when the water supply is limited [19].

After a period of high precipitation in February and March (433 mm), the soil water content at depths greater than 40 cm was higher in the monoculture pasture than in the integrated system until May (Figures 2 and 3). In the dry months of April, May, and June, only 46 mm of rainfall was observed (Figure 1D). As the drought progressed, the soil water content in the monoculture pasture decreased and became similar to that in the agroforestry system (Figures 2 and 3) due to the higher evapotranspiration, solar radiation, and wind speed in the pasture [13]. After the rainy period, the soil water content at depths greater than 40 cm decreased in the agroforestry system as a consequence of the *Eucalyptus* root system associated with grass roots. The high water use of *Eucalyptus* has led to concerns about the environmental impacts of its cultivation [34]. However, some improved *Eucalyptus* hybrids have efficient water use characteristics and are more suitable for semi-arid regions [41].

At soil depths greater than 50–60 cm, there was little variation in the soil water content with distance from the trees (Figure 4). According to Bosi et al. [13], in the superficial layer (0–30 cm), soil water availability is more dynamic due to faster water uptake by plants and evaporation from the soil in the first days of a dry spell. Roots are strongly concentrated in surface horizons; hence, greater competition for water occurs in the superficial layers of the soil [35]. This root distribution is due to the higher concentration of nutrients in the upper layer, which can be improved by the cycling of nutrients [42]. Bouillet et al. [43] observed that up to 53% of fine *Eucalyptus* roots are concentrated in the 0–25 cm layer. In the present study, the soil volumetric water content in the topsoil was expected to increase with distance from the tree row due to greater competition between the trees and companion crop for in-row resources, whereas this competition should decrease with distance from the tree row. Continued research on long-term effects is necessary to develop effective strategies to minimize adverse tree–crop interactions, meet production objectives, and promote environmental benefits.

*Forage Biomass*

The trees introduced into the pastures prompted changes in the soil water content and consequently forage production (Figure 5). The addition of *Eucalyptus* close to the trees up to 2.0 m reduced biomass forage production. The reduction in forage production near the rows of trees was due to the reduction in soil moisture caused by the greater activity of the root system [3–10]. Besides the water and nutrients' effects, forage production is impacted by shade intensity [21]. Studies have shown that forage growth is reduced at shading above 35–40% [44]. Bosi et al. [17] reported physiological changes in the grass canopy at 2 m from

the tree groves. Vieira Junior et al. [10] observed that in row spacing of 30 and 45 m, the availability of radiation did not limit forage production, while the agroforestry system with 15 m of row spacing required more attention for its management.

The greater competition for water and lower forage production in the agroforestry system compared to the monoculture pasture in April and October 2015 are due to the agroforestry system's potential to exploit water resources, particularly soil water captured by the trees [45]. Soil water dynamics in an agroforestry system depend on the region's rainfall and agroforestry system configuration [46]. The forage biomass was lower in October 2014 because sampling occurred after a period of reduced soil water content due to an accumulated rainfall of 44 mm in October. The soil water content was lower in October 2014 than in April 2014 and 2015 (Figures 2 and 3), due to the lower rainfall in the previous period, with monthly precipitation of 10.8 mm in May, 3.1 mm in June, 53.0 mm in July, 0.0 mm in August, and 101.9 mm in September of 2014 (Figures 2 and 3). The soil water content in October 2014 was 7.02% with little difference between soil depths. In addition, the forage biomass was clearly decreased by the introduction of *Eucalyptus* in the system, and this effect was greater during months of higher soil water content and higher forage biomass production, such as April 2014 and 2015 and October 2015 (Figures 2 and 3). The competition may be higher due to the low fertility of tropical sandy soils [47]. Sandy soils have low water storage capacity, which can limit the water supply and increase competition with forage grasses [48,49]. This is a limiting factor, particularly in poor soils such as those in the Cerrado region or the African savannas [50].

Despite the decreases in the soil water content near the *Eucalyptus* row and in forage production, this agroforestry system has the potential for carbon sequestration [51]. Sarto et al. [52] observed greater carbon sequestration by plant biomass in this agroforestry system compared with the monoculture pasture, mainly by the forest component. Trees increase the potential capacity of integrated systems to remove atmospheric carbon and mitigate greenhouse gas emissions [53]. The most important factor for fostering the storage of soil organic carbon in agroforestry systems is increasing the land equivalent ratio [54] through the vertical and temporal integration of crops and trees and more effective use of sunlight, water, and nutrients [55]. In addition to carbon sequestration, agroforestry systems are efficient land use management systems for food production due to the greater thermal comfort provided by trees to animals, especially in the hottest months of the year [56]. Moreover, harvesting wood may generate additional income for farmers, which is important from the standpoint of economic sustainability [45].

Agricultural production diversification has the advantage in agroforestry compared with pasture [57]. A higher total aboveground biomass yield in agroforestry systems has been observed, indicating better land use [36]. However, competition between trees and pastures may be the limiting factor if the system is not managed well [46].

## 5. Conclusions

Under tropical conditions and coarse-textured soils, *Eucalyptus* trees altered the soil water content in an agroforestry system compared with a monoculture pasture. The soil water content within the top 40 cm of the soil profile increased in the agroforestry system with increasing distance from the trees in the months with higher rainfall. In the dry season, the soil water content within the soil profile (up to 100 cm) was similar between the *Eucalyptus* inter-rows (i.e., at 2.0, 4.0, and 6.0 m from the tree rows) and monoculture pasture. Overall, this study showed that introducing *Eucalyptus* into a pasture (agroforestry system) can decrease the soil water content, especially in the superficial layers. In most seasons, the agroforest systems reduced the forage production close to the *Eucalyptus* tree rows, up to 2 m from the trees, due to the lower soil water content in the tropical system. Therefore, in tropical regions with sandy soils, the competition of grass and trees should be considered in the establishment of the integrated system to enhance the potential advantages and benefits of the diversified agroecosystem.

**Author Contributions:** M.V.M.S., W.L.B.B. and C.A.R.: investigation, methodology, data curation, formal analysis, writing—original draft, writing—review and editing. C.A.R.: conceptualization, project administration, resources, supervision, writing—original draft, writing—review and editing. W.L.B.B., D.B., M.R.N. and C.W.R.: writing—original draft, writing—review and editing. All authors have read and agreed to the published version of the manuscript.

**Funding:** São Paulo Research Foundation, FAPESP [grants #2013/08664-5, #2014/10656-3, and #2016/14323-4]; and the Brazilian National Council for Scientific and Technological Development, CNPq.

**Data Availability Statement:** The data that support the findings of this study are available in the main body of the paper. Raw data can also be obtained (on request) from the corresponding author.

**Acknowledgments:** The authors acknowledge support from the Coordination for Improvement of Higher Education Personnel, CAPES.

**Conflicts of Interest:** The authors declare no conflict of interest.

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
