# Peer review of "Deep Soil Water Content and Forage Production in a Tropical Agroforestry System"

_agriculture, doi:10.3390/agriculture12030359_

Round 1
Reviewer 1 Report
I do find this work interesting and valuable, but there are some issues which needs explanation.
Lines 162 – 163 - what does this sentence refer to? One sentence earlier you have mentioned that minimum average soil water content in dry season was 8.2% in the 0-10 cm layer.
Line 161 – here you have mantioned soil layers deeper than 50 – 60 cm, so the reference in the brackets should be only to Figure 3.
Line 165 – and what about 0.0 m? From Figure 2E and 3A-E it seems that it applies also to 0.0m. The same for line 159.
Lines 166 – 168 – I do not agree with that statement. Looking at Figures 2D and 2E there are differences especially in comparision to 0.0m. I would rather say that it is true at depths greater than 50-60cm.
Line 170 – but Figure 4 does not refer to months with higher rainfall.
Line 172 – it is not true – according to the Figure 4 in the 90 – 100 cm layer there aren’t any differences between eucalyptus row (0.0 m) and pasture, so why you mentioned here that difference increased to 9%?
Line 209 - Grevillea robusta should be in italic
Author Response
Reviewer 1 Comments and Suggestions for Authors
I do find this work interesting and valuable, but there are some issues which needs explanation.
Lines 162 – 163 - what does this sentence refer to? One sentence earlier you have mentioned that minimum average soil water content in dry season was 8.2% in the 0-10 cm layer.
Reviewed.
The soil water content in the 0−10 cm layer ranged from 6.3% to 16.5% in the dry season. However, we used only the average values.
Reviewer 2 suggested changes to the results section. Therefore, some sentences were deleted and rewritten to be clear.
Line 161 – here you have mantioned soil layers deeper than 50 – 60 cm, so the reference in the brackets should be only to Figure 3.
Reviewed.
Line 160 – 164 - From April to September (dry season), the soil water content in the Eucalyptus inter-rows (ie, at 0.0, 2.0, 4.0, and 6.0 m) was similar (p > 0.05) to that in the monoculture pasture in all layers (Figures 2 and 3). In these months, the minimum average soil water content was 7.6% in soil layers deeper than 50–60 cm (Figure 2) and 8.2% in the 0–10 cm layer (Figure 2A).
Line 165 – and what about 0.0 m? From Figure 2E and 3A-E it seems that it applies also to 0.0m. The same for line 159.
Reviewed.
Line 168-172- In May 2014 (dry season), at depths greater than 40–50 cm, soil water content was higher significantly (p < 0.05) in the monoculture pasture than in the Eucalyptus inter-rows (ie, 0.0, 2.0, 4.0, and 6.0 m; Figures 2E, 3A–E). In the rainy seasons, the variation in soil water content at depths greater than 50–60 cm was small (p < 0.05) in the Eucalyptus transect (0.0, 2.0, 4.0, and 6.0 m) and monoculture pasture (Figures 2 and 3).
Lines 166 – 168 – I do not agree with that statement. Looking at Figures 2D and 2E there are differences especially in comparision to 0.0m. I would rather say that it is true at depths greater than 50-60cm.
Reviewed.
Line 160-164 - From April to September (dry season), the soil water content in the Eucalyptus inter-rows (ie, at 0.0, 2.0, 4.0, and 6.0 m) was similar (p > 0.05) to that in the monoculture pasture in all layers (Figures 2 and 3). In these months, the minimum average soil water content was 7.6% in soil layers deeper than 50–60 cm (Figure 2) and 8.2% in the 0–10 cm layer (Figure 2A).
Line 170 – but Figure 4 does not refer to months with higher rainfall.
Reviewed.
Line 172-175 - In general, soil water content increased significantly (p < 0.05) with distance from the trees in soil layers up to a depth of 30–40 cm (Figure 4). Averaged across months, soil water content in the 30–40 cm layer was 7.2% lower (p < 0.05) under the Eucalyptus tree row than in the monoculture pasture (Figure 4).
Line 172 – it is not true – according to the Figure 4 in the 90 – 100 cm layer there aren’t any differences between eucalyptus row (0.0 m) and pasture, so why you mentioned here that difference increased to 9%?
The sentence was incorrect. It was reviewed.
In general, soil water content increased significantly (p < 0.05) with distance from the trees in soil layers up to a depth of 30–40 cm (Figure 4). Averaged across months, soil water content in the 30–40 cm layer was 7.2% lower (p < 0.05) under the Eucalyptus tree row than in the monoculture pasture (Figure 4).
Line 209 - Grevillea robusta should be in italic
Reviewed.
In general, soil water content was lower in the Eucalyptus tree rows than in the monoculture pasture throughout the year, with greater differences in months with higher rainfall (Figures 2 and 3). This difference in soil water content reflects interspecific competition. Competition for light and/or water has been reported to decrease the yields of some species [30, 31]. The tropical sandy soil examined in the present study has low water storage capacity, increasing the importance of competition for water. The lower soil water content within the Eucalyptus tree rows reflects higher water uptake by plants due to the concentration of fine roots of Eucalyptus [32]. Radersma and Ong [33] observed decreases in soil water content of 2–7% (vol. basis) near fast-growing species such as E. grandis and Grevillea robusta and linked these decreases to greater water use by Eucalyptus and Grevillea compared with Cedrella and Markhami in a Rhodic Acrudox with 58% clay. Prasad et al. [34] also attributed low water content at depths of up to 40 cm in the proximity of tree rows to water absorption by the trees.
Please let us know if there is anything that we should review.
Thank you,
Reviewer 2 Report
This study examines the influence of the introduction of Eucalyptus trees in a pasture on soil water. Studies like this are very important from ecohydrological perspective. The authors put a lot of effort to collect the soil moisture data. However, I think that this manuscript can be further improved. Some methods are not described very well, which makes the interpretation of the results difficult. More comments and suggestions are given below:
Section 2.1: Is there any information about reference evapotranspiration? How was rainfall and temperature measured? Was a meteorological station installed nearby the site? Is the pasture at the agroforestry system the same as the monoculture pasture?
Section 2.2: Please state the study period. Also, is there any information about the maximum rooting depth of the eucalyptus trees and of the pasture?
Lines 100 – 105: This sentence is not very clear. Better to rephrase it
Line 108: Did Eucalyptus trees grow over 12m in two years?
Lines 112 – 118: How is this information related to this study?
Table 1: Do we need the chemical characteristics of the soil?
Section 2.4: What was the measuring interval of the soil moisture? What is the accuracy of this sensor and what are the units of measurements? Did the authors measured soil moisture from all sites on the same days?
Line 146: Is the rainy season from October to April? Or from October to June?
Section: I think the authors need to re-arrange the results in a more logic order, so the reader can compare the soil moisture results from the monoculture pasture with the results of the Agroforestry system for each depth, season and distance from tree. For instance, there is no connection between the sentences in lines 146 – 151. Also, there are no results on the statistical analysis described in section 2.6.
Figure 4: I would expect higher soil moisture in the deeper soil layers than in the shallow soil layers due drainage and lower evaporation. Or not? Do the roots of pasture and eucalyptus reach the 1 m soil depth?
Lines 201 – 203: Is the 4.3% difference statistical significant? Can this difference be due to soil heterogeneity?
Lines 245 – 247: Is this the case for Eucalyptus species too?
Author Response
Reviewer 2 Comments and Suggestions for Authors
This study examines the influence of the introduction of Eucalyptus trees in a pasture on soil water. Studies like this are very important from ecohydrological perspective. The authors put a lot of effort to collect the soil moisture data. However, I think that this manuscript can be further improved. Some methods are not described very well, which makes the interpretation of the results difficult. More comments and suggestions are given below:
Section 2.1: Is there any information about reference evapotranspiration? How was rainfall and temperature measured? Was a meteorological station installed nearby the site? Is the pasture at the agroforestry system the same as the monoculture pasture?
The precipitation was determined by a meteorological station close to the experiment. But we don't have ETO data. The pasture was similar to the agroforestry system and the monoculture pasture.
Section 2.2: Please state the study period. Also, is there any information about the maximum rooting depth of the eucalyptus trees and of the pasture?
Reviewed.
The experimental design was completely randomized with six replicates from October 2013 to June 2015.
Lines 100 – 105: This sentence is not very clear. Better to rephrase it
Reviewed.
The experiment was carried out in a 10-year degraded pasture of approximately 6 ha. The area was prepared by plowing and disking, and 2,000 kg ha–1 of limestone applied to raise the soil base saturation to 60%. In October 2009, 200 kg P ha–1 of phosphorus was applied using reactive rock phosphate. Eucalyptus [hybrid E. urograndis H–13 (Eucalyptus urophylla × E. grandis) and hybrid E. grancam 1277 (E. grandis × E. camaldulensis)] were planted east-west with 12.0 m distance between rows and 2.0 m between plants using 3-month-old seedlings at a density of 370 plants ha–1. Palisade grass (Urochloa brizantha cv. Marandu) was sown 8.0 cm deep at a density of 7 kg ha–1 (pure live seed = 76%) using a no-till drill at a row spacing of 0.34 m.
Line 108: Did Eucalyptus trees grow over 12m in two years?
Yes, Eucalyptus are much faster growing than most other trees in cultivation. Many eucalyptus types grow as fast as 10 to 15 feet per year.
Lines 112 – 118: How is this information related to this study?
We believe that this area management information is important. This information demonstrates pasture management.
Table 1: Do we need the chemical characteristics of the soil?
Yes, the soil fertility can affect pasture and eucalyptus root growth at depth.
Section 2.4: What was the measuring interval of the soil moisture? What is the accuracy of this sensor and what are the units of measurements? Did the authors measured soil moisture from all sites on the same days?
On average, the measuring interval was every 25 days. The sensor exhibited measurement precision better than 0,01 m3 m-3. The measurements were executed on the same day, between 9 and 11 a.m.
Line 146: Is the rainy season from October to April? Or from October to June?
Reviewed.
From October to April 2013/2014 and from October to April 2014/2015 (rainy seasons), soil water content was lowest close to the Eucalyptus tree rows, i.e., at 0.0 and 2.0 m, in the 0−10 cm (Figure 2A), 10−20 cm (Figure 2B), 20−30 cm (Figure 2C), 30−40 cm (Figure 2D) and 40−50 cm (Figure 2E) soil layers.
Section: I think the authors need to re-arrange the results in a more logic order, so the reader can compare the soil moisture results from the monoculture pasture with the results of the Agroforestry system for each depth, season and distance from tree. For instance, there is no connection between the sentences in lines 146 – 151. Also, there are no results on the statistical analysis described in section 2.6.
Reviewed.
Results have been improved.
The statistic was added in the data presentation.
Some sentences have been rearranged.
Figure 4: I would expect higher soil moisture in the deeper soil layers than in the shallow soil layers due drainage and lower evaporation. Or not? Do the roots of pasture and eucalyptus reach the 1 m soil depth?
Reviewed.
Yes, eucalyptus roots and grass roots consume water at depth. This fact is linked to the characteristics of sandy soil as well, to store little water. Please find more information on Sarto, M. V.; Borges, W. L.; Sarto, J. R.; Rice, C. W.; Rosolem, C. A. Root and shoot interactions in a tropical integrated crop–livestock–forest system. Agrofor. Syst., 2020c, 181, 102796. https://doi.org/10.1016/j.agsy.2020.102796
Lines 201 – 203: Is the 4.3% difference statistical significant? Can this difference be due to soil heterogeneity?
Corrected.
In general, soil water content was lower in the Eucalyptus tree rows than in the monoculture pasture throughout the year, with greater differences in months with higher rainfall.
Lines 245 – 247: Is this the case for Eucalyptus species too?
Yes
At soil depths greater than 50–60 cm, there was little variation in soil water content with distance from the trees (Figure 4). According to Bosi et al. [13], in the superficial layer (0–30 cm), soil water availability is more dynamic due to faster water uptake by plants and evaporation from the soil in the first days of a dry spell. Roots are strongly concentrated in surface horizons, and hence greater competition for water occurs in the superficial layers of the soil [35]. This root distribution is due to the higher concentration of nutrients in the upper layer, which can be improved by cycling of nutrients [41]. Bouillet et al. [42] observed that up to 53% of fine Eucalyptus roots are concentrated in the 0–25 cm layer. In the present study, soil volumetric water content in the topsoil was expected to increase with distance from the tree row due to greater competition between the trees and companion crop for in-row resources, whereas this competition should decrease with distance from the tree row. Continued research on long-term effects is necessary to develop effective strategies to minimize adverse tree-crop interactions, meet production objectives and promote environmental benefits.
Please let us know if there is anything that we should review.
Thank you,
Reviewer 3 Report
The article actually has a balance sheet and the totals are presented as bar charts. Little has been written about energy efficiency, for example. This is important in order to lay the foundations for research on energy efficiency. This shows how the emission of emission gases is balanced. I recommend that the authors read the article https://doi.org/10.3390/en14175574
There is no in-depth statistical analysis. Statistical analysis was not sufficiently discussed. The entire manuscript looks like an excerpt from a report from some major work. Without statistics and research, the article does not contribute to any knowledge development. An expansion of literature, conclusions and discussions is required.
Author Response
Revisor 3 Comments and Suggestions for Authors
The article actually has a balance sheet and the totals are presented as bar charts. Little has been written about energy efficiency, for example. This is important in order to lay the foundations for research on energy efficiency. This shows how the emission of emission gases is balanced. I recommend that the authors read the article https://doi.org/10.3390/en14175574
Energy efficiency and GHG emissions are important. However, our article evaluated the impact of introducing Eucalyptus trees into a pasture on soil water content throughout the soil profile. Soil moisture was sampled at 0.0 (planting row), 2.0, 4.0, and 6.0 m (midpoint between tree rows) from the Eucalyptus tree rows. A total of 8,820 readings were made from October 1, 2013 to June 1, 2015.
There is no in-depth statistical analysis. Statistical analysis was not sufficiently discussed. The entire manuscript looks like an excerpt from a report from some major work. Without statistics and research, the article does not contribute to any knowledge development. An expansion of literature, conclusions and discussions is required.
The data were tested for normality, residual independence, and homogeneity of variance. When these requirements were met, the data were submitted to one-way. If significant differences (p < 0.05) were detected by ANOVA, individual comparisons were made using the t test (p < 0.05). Means were compared via t test (Bars in the graphics). Bars indicate LSD (0.05) values for dates with significant differences between sampling locations
Forage biomass and soil volumetric water on different dates (October 1, 2013 to June 1, 2015) were considered to be fixed effects, and blocks were considered to be random effects.
In figure 5, the different letters represent the statistical difference. Means for forage in each season followed by the same letters do not differ significantly by the t-test (LSD), 5% probability.
Please let us know if there is anything that we should review.
Thank you,
Round 2
Reviewer 2 Report
The authors did many improvements in the revised version of the manuscript. However, there are few comments/suggestions that were not addressed properly. More details below:
The authors responded to my previous question in Section 2.1 that “The pasture was similar to the agroforestry system and the monoculture pasture.” What do you mean by “similar”? Can you provide more information in the manuscript?
I didn’t get any reply in my question about the maximum rooting depth of the eucalyptus trees and of the pasture.
The authors responded to my question in section 2.4 “On average, the measuring interval was every 25 days. The sensor exhibited measurement precision better than 0,01 m3 m-3. The measurements were executed on the same day, between 9 and 11 a.m.” However, I didn’t find anything written in the manuscript.
Section 2.6 and Section 3.1: Please state clearly the statistical tests performed and the hypotheses stated so we can understand the meaning of the p-value. The way it is written now is confusing.
General comments:
The responses should provide also the line numbers where the changes done in the revised manuscript.
Some of the changes done in the revised version were highlighted but some not.
Author Response
Review 2
The authors did many improvements in the revised version of the manuscript. However, there are few comments/suggestions that were not addressed properly. More details below:
The authors responded to my previous question in Section 2.1 that “The pasture was similar to the agroforestry system and the monoculture pasture.” What do you mean by “similar”? Can you provide more information in the manuscript?
Line 93-95: In addition, a monoculture pasture in the same soil and managed in the same way as ICLFS was used as reference (Figure 1C).
- Monoculture pasture in the same soil and managed in the same way as Agroforestry system;
- Monoculture pasture close to the agroforestry, it was in line with the experimental design
I didn’t get any reply in my question about the maximum rooting depth of the eucalyptus trees and of the pasture.
Line 131-133: Soil water content was measured every 0.1 m to a depth of 1.0 m, the area with the greatest presence of roots. From 1.0 m the fine roots decreased gradually [30].
In the literature, root gowth of eucalyptus and pasture was generally evaluated up to 1.0 m, Bosi et al. [13], Sarto et al. [52]
The authors responded to my question in section 2.4 “On average, the measuring interval was every 25 days. The sensor exhibited measurement precision better than 0,01 m3 m-3. The measurements were executed on the same day, between 9 and 11 a.m.” However, I didn’t find anything written in the manuscript.
Line 129-130: On average, the measuring interval was every 25 days. The measurements were executed on the same day, between 9 and 11 a.m
Line 137-138: The sensor exhibited measurement precision better than 0.01 m3 m-3.
Section 2.6 and Section 3.1: Please state clearly the statistical tests performed and the hypotheses stated so we can understand the meaning of the p-value. The way it is written now is confusing.
Line 149-153: The experimental statistic has been improved
The data were checked for normality (Shapiro–Wilk test), the independence of residuals, and the variance of homogeneity (Levene’s test). All statistical analyses were performed by software Sisvar® (Statistical Analysis Software, UFLa, Lavras, MG, Brazil). The dataset of soil water was submitted to ANOVA for each month and means were compared by Fisher’s protected LSD test (p < 0.05). The forage production was statistically significant at p < 0.05, by Fisher's protected LSD test (p < 0.05).
Thank you,